# NLProlog: Reasoning with Weak Unification for Natural Language Question Answering

## Abstract

Symbolic logic allows practitioners to build systems that perform rule-based reasoning which is interpretable and which can easily be augmented with prior knowledge. However, such systems are traditionally difficult to apply to problems involving natural language due to the large linguistic variability of language. Currently, most work in natural language processing focuses on neural networks which learn distributed representations of words and their composition, thereby performing well in the presence of large linguistic variability. We propose to reap the benefits of both approaches by applying a combination of neural networks and logic programming to natural language question answering. We propose to employ an external, non-differentiable Prolog prover which utilizes a similarity function over pretrained sentence encoders. We fine-tune these representations via Evolution Strategies with the goal of multi-hop reasoning on natural language. This allows us to create a system that can apply rule-based reasoning to natural language and induce domain-specific natural language rules from training data. We evaluate the proposed system on two different question answering tasks, showing that it complements two very strong baselines – BIDAF (Seo et al., 2016a) and FastQA (Weissenborn et al., 2017) – and outperforms both when used in an ensemble.

## 1 Introduction

We consider the problem of multi-hop reasoning on natural language input. For instance, consider the statements *Socrates was born in Athens* and *Athens belongs to Greece*, together with the question *Where was Socrates born?* There are two obvious answers following from the given statements: *Athens* and *Greece*. While *Athens* follows directly from the single statement *Socrates was born in Athens*, deducing *Greece* requires a reader to combine both provided statements using the knowledge that *a person that was born in a city, which is part of a country, was also born in the respective country*.

Most recent work that addresses such challenges leverages deep learning based methods (Sukhbaatar et al., 2015; Peng et al., 2015; Seo et al., 2016b; Raison et al., 2018; Henaff et al., 2016; Kumar et al., 2016; Graves et al., 2016; Dhingra et al., 2018), capable of dealing with the linguistic variability and ambiguity of natural language text. However, the black-box nature of neural networks makes it hard to interpret the exact reasoning steps leading to a prediction (local interpretation), as well as the induced model (global interpretation).

Logic programming languages like Prolog (Wielemaker et al., 2012), on the other hand, are built on the idea of using symbolic rules to reason about entities, which makes them highly interpretable both locally and globally. The capability to use user-defined logic rules allows users to incorporate external knowledge in a straightforward manner. Unfortunately, because of their reliance on symbolic logic, systems built on logic programming need extensive preprocessing to account for the linguistic variability that comes with natural language (Moldovan et al., 2003).

We introduce NLProlog, a system which combines a symbolic reasoner and a rule-learning method with pretrained sentence representations to perform rule-based multi-hop reasoning on natural language input.[1] Like inductive logic programming methods, it facilitates both global as well as

---

[1] NLProlog and our evaluation code will be made open-source upon publication.

local interpretation, and allows for straightforward integration of prior knowledge. Similarly to deep learning based approaches, it can be applied to natural language text without the need to transforming it to formal logic.

At the core of the proposed method is an external non-differentiable theorem prover which can take similarities between symbols into account. Specifically, we modify a Prolog interpreter to support weak-unification as proposed by Sessa (2002). To obtain similarities between symbols, we utilize sentence encoders initialized with pretrained sentence embeddings (Pagliardini et al., 2017) and then fine-tune these for a downstream question answering task via gradient-based optimization methods. Since the resulting system contains non-differentiable components, we propose using Evolution Strategies (ES) (Eiben et al., 2003) as a gradient estimator (Williams, 1992) for training the system – enabling us to fine-tune the sentence encoders and to learn domain-specific logic rules (e.g. that the relation *is in* is transitive) from natural language training data. This results in a system where training can be trivially parallelized, and which allows to change the logic formalism by simply exchanging the external prover without the need for an intricate re-implementation as an end-to-end differentiable function.

In summary, our main contributions are: *a)* we show how Prolog-like reasoning can be applied to natural language input by employing a combination of pretrained sentence embeddings, an external logic prover, and fine-tuning using Evolution Strategies, *b)* we extend a Prolog interpreter with weak unification based on distributed representations, *c)* we present Gradual Rule Learning (GRL), a training algorithm that allows the proposed system to learn First-Order Logic (FOL) rules from entailment, and *d)* we evaluate the proposed system on two different Question Answering (QA) datasets and demonstrate that its performance is on par with state-of-the-art neural QA models in many cases, while having different failure modes. This allows to build an ensemble of NLPROLOG and a neural QA model that outperforms all individual models.

## 2    RELATED WORK

Our work touches in general on weak-unification based fuzzy logic (Sessa, 2002) and focuses on multi-hop reasoning for QA, the combination of logic and distributed representations, and theorem proving for question answering.

**Multi-hop Reasoning for QA.**    One prominent approach for enabling multi-hop reasoning in neural QA models is to iteratively update a query embedding by integrating information from embeddings of context sentences, usually using an attention mechanism and some form of recurrency (Sukhbaatar et al., 2015; Peng et al., 2015; Seo et al., 2016b; Raison et al., 2018). These models have achieved state-of-the-art results in a number of reasoning-focused QA tasks. Henaff et al. (2016) employ a differentiable memory structure that is updated each time a new piece of information (usually a sentence) is processed. The memory slots can be used to track the state of various entities, which can be considered as a form of temporal reasoning. Similarly, the Neural Turing Machine (Graves et al., 2016) and the Dynamic Memory Network (Kumar et al., 2016), which are built on differentiable memory structures, have been used to solve synthetic QA problems requiring multi-hop reasoning. Dhingra et al. (2018) modify an existing neural QA model to additionally incorporate coreference information provided by a coreference resolution model in a preprocessing step, which improves performance on QA problems requiring multi-hop reasoning.

All of the methods above perform reasoning more or less implicitly by updating latent vector representations, which makes an unambiguous interpretation of the exact reasoning steps difficult. Additionally, it is not obvious how a strong prior, like user-defined inference rules, could be imposed on the respective reasoning procedures. Besides, many of them have been evaluated only on artificially generated data sets and thus it is unclear how they perform when on data that involves natural linguistic variability.

**Combination of FOL and Distributed Representations.**    Investigating the combination of formal logic and distributed representations has a long tradition, which is reviewed by Besold et al. (2017). Strongly related to our approach is the combination of Markov Logic Networks (Richardson & Domingos, 2006), Probabilistic Soft Logic (Bach et al., 2017), and word embeddings, which has been applied to Recognizing Textual Entailment (RTE) and Semantic Textual Similarity (STS) (Garrette

et al., 2011; 2014; Beltagy et al., 2013; 2014), and improves upon baselines utilizing either only logic or only distributed representations.

An area in which neural multi-hop reasoning models have been thoroughly investigated is Knowledge Base Completion (Das et al., 2016; Cohen, 2016; Neelakantan et al., 2015; Das et al., 2017). While QA could be in principle modeled as a KB-completion task, the construction of a densely connected KB from text is far from trivial, due to the inherent ambiguity of natural language. Without any preprocessing, even the moderately sized QA tasks considered in this work would produce a very large and sparsely connected KB.

Closest to our approach is the Natural Theorem Prover (NTP) (Rocktäschel & Riedel, 2017), which obtains the final proof score for a statement by constructing a neural network that represents all possible proofs. The model is trained end-to-end using backward chaining and a differentiable unification operator. Since the number of possible proofs grows exponentially with the number of facts and rules, NTPs cannot scale even to moderately sized knowledge bases, and are thus not applicable to natural language problems in its current form. We circumvent this issue by using a non-differentiable prover and fine-tune the model using Evolution Strategies.

**Theorem Proving for Question Answering.**  Our work is not the first to apply theorem proving to QA problems. Angeli et al. (2016) employ a system based on Natural Logic to search a large KB for a single statement that entails the candidate answer. This is somewhat orthogonal to our approach, as we aim to learn rules that combine multiple statements to answer a question.

More traditional systems like Watson (Ferrucci et al., 2010) or COGEX (Moldovan et al., 2003) utilize an integrated theorem prover, but require a transformation of the natural language input to logical form. In the case of COGEX, this improves the accuracy of the underlying system by 30%, and increases its interpretability. While this work is similar in spirit, we greatly simplify the preprocessing step by replacing the transformation of natural language to logic with the simpler approach of taking Open Information Extraction (Open IE) (Etzioni et al., 2008) textual patterns.

Fader et al. (2014) propose the OPENQA system that utilizes a mixture of handwritten and automatically obtained operators that are able to parse, paraphrase and rewrite queries, which allows them to perform large-scale QA on KBs that include Open IE triples. While this work shares the same goal – answering questions using facts extracted by Open IE – we choose a completely different approach to address the problem of linguistic variablity and focus on the combination of multiple facts by learning logical rules.

## 3 BACKGROUND

In the following, we review the background relevant for the custom Prolog engine employed by our method. Specifically, we briefly introduce Prolog's backward chaining algorithm and unification procedure (Russell & Norvig, 2016). We assume basic knowledge of formal logic and logic programming.

In a nutshell, a Prolog program consists of a set of rules in the form of Horn clauses

$$h(f_1^h, \ldots, f_n^h) \ \Leftarrow \ p_1(f_1^1, \ldots, f_m^1) \ \wedge \ \ldots \ \wedge \ p_B(f_1^B, \ldots, f_l^B),$$

where $h, p_i$ are predicate symbols and $f_j^i$ are either function (denoted in lower case) or variable (upper case) symbols. The domain of function symbols is denoted by $\mathcal{F}$ and the domain of predicate symbols by $\mathcal{P}$. $h(f_1^h, \ldots, f_n^h)$ is called the *head* and $p_1(f_1^1, \ldots, f_m^1) \ \wedge \ \ldots \ \wedge \ p_B(f_1^B, \ldots, f_l^B)$ the *body* of the rule. We call $B$ the *body size* of the rule and rules with a body size of zero are named *atoms* (short for *atomic formula*).[2] If an atom does not contain any variable symbols it is termed *fact*. As in related work (Sessa, 2002; Julián-Iranzo et al., 2009), we disregard negation and disjunction.

The central procedure of Prolog is *unification*. It receives two atoms and attempts to find variable substitutions that make both atoms syntactically equal. For example, the input atoms *country*(*Greece, Socrates*) and *country*(*X, Y*) result in the following variable substitution after unification: $\{X/Greece, Y/Socrates\}$.

---

[2]For simplicity, we only consider function-free Prolog in our experiments, i.e., Datalog (Gallaire & Minker, 1978) programs where all function symbols have arity zero and are called *entities*. However, in principle NLPROLOG also supports functions with higher arity.

The proof algorithm of Prolog is called *backward-chaining*. It starts from a goal atom $g$ and attempts to prove it by applying suitable rules, thereby generating subgoals that are proved next. To find applicable rules, it attempts to unify $g$ with the heads of all available rules. If this unification succeeds, the resulting variable substitutions are applied to the atoms in the rule body and each of those atoms becomes a new subgoal. For instance, the application of the rule $country(X, Y) \Leftarrow born\_in(Y, X)$ to the goal $country(Greece, Socrates)$ would yield the subgoal $born\_in(Socrates, Greece)$. Then the process is repeated for all subgoals until no subgoal is left, i.e., until all subgoals have been unified with a fact. The result of this procedure is a set of rule applications and variable substitutions called proof. Note, that the number of possible proofs is exponential in the number of predicate and entity symbols, as every rule might be used in the proof of each subgoal. Pseudo code for weak unification can be found in Appendix A.1 and we refer the reader to Russell & Norvig (2010) for more details. To apply standard Prolog to a NLP problem like QA, one would have to account for semantic similarities and ambiguities with extensive and error-prone preprocessing, e.g. when transforming the natural language input to logical form.

## 4 NLPROLOG

Our aim is to apply Prolog to natural language question answering where the same entity or relation can have different natural language surface forms. Thus, we replace the equality-based unification operator with similarity-based weak unification (Sessa, 2002), which allows to unify two symbols $x, y$ if they are sufficiently similar, as judged by a similarity function $\sim_\theta$ parameterized by $\theta$. Then, the result of unification also contains a proof success score $S$ that is the result of the symbols' similarity and the previous success score $S'$: $S = \top(x \sim y, S')$, where $\top \in \{\min, \cdot\}$ is an aggregation function. The result of backward-chaining with weak unification are (possibly) multiple proofs, each with an associated proof success score. NLPROLOG combines inference based on weak unification and distributed representations to allow reasoning on natural language statements. The natural language statements are first transformed to triples using Open IE (Etzioni et al., 2008). The symbols occurring in these triples and in the rules are then embedded into a vector space, which in turn is used to estimate similarities between symbols. The resulting similarity function is subsequently used to perform a proof and consequently obtain a proof success score $S$. The proof success score is then utilized as a training signal for ES. An illustration of the process can be found in Fig. 1, where we visualize the interplay of the different components for our running example.

### 4.1 SIMILARITY COMPUTATION

We embed symbols using an encoder $\mathcal{E}_\theta : \mathcal{F} \cup \mathcal{P} \mapsto \mathbb{R}^d$ parametrized by $\theta$ for entity and predicate symbols, where $d$ denotes the embedding size. The resulting embeddings are used to induce $\sim_\theta$: $(\mathcal{F} \cup \mathcal{P})^2 \mapsto \mathbb{R}$:

$$x \sim_\theta y = \frac{1 + cos(\mathcal{E}_\theta(x), \mathcal{E}_\theta(y))}{2} \qquad (1)$$

where *cos* denotes the cosine similarity between two vectors. There are alternative similarity functions such as Euclidean distance or RBF kernel, but in preliminary experiments we found cosine simlarity to work more robustly.

We use an encoder function that uses a embedding lookup table for predicate symbols and a different one for entities. All embeddings representing natural language phrases are populated with sentence vectors that were pretrained with SENT2VEC (Pagliardini et al., 2017). Additionally, we introduce a third lookup table for the predicate symbols of rules and goals, because semantics of goal or rule predicates might differ from the semantics of fact predicates even if they share the same surface form. For instance, the query $(X, parent, Y)$ could be interpreted either as $(X, is\ the\ parent\ of, Y)$ or as $(X, has\ the\ parent, Y)$ which are fundamentally different things.

### 4.2 FINE-TUNING THE ENCODER TO A DOWNSTREAM TASK

We propose to fine-tune the similarity function on a downstream task by updating the symbol embeddings. As NLPROLOG involves the non-differentiable proof search step, we cannot apply backpropagation for optimization. Instead, we propose to employ Evolution Strategies in conjunction with Adam (Kingma & Ba, 2014) to estimate the weight updates. ES recently showed good results

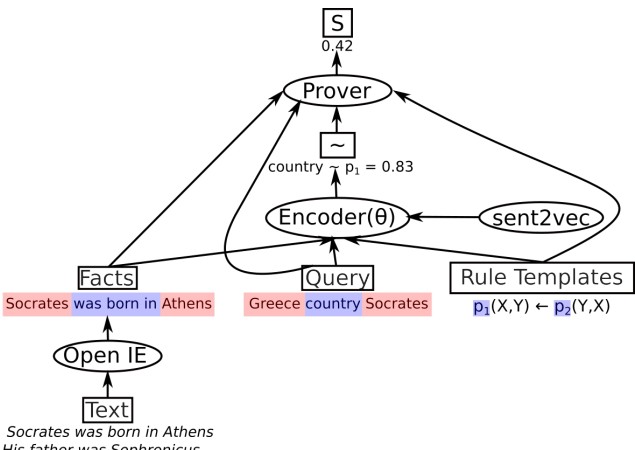

Figure 1: Overview of NLPROLOG. The components of NLPROLOG are depicted as ellipses, while inputs and outputs are drawn as squares. Phrases with red background are entities and blue ones are predicates.

for Reinforcement Learning problems in Salimans et al. (2017); Mania et al. (2018). More formally, the parameter update is computed as:

$$\theta_{t+1} = \theta_t + \frac{\alpha_t}{N\sigma_J^t} \sum_{\substack{k=1 \\ \epsilon_k \sim \mathcal{N}(0,\sigma)}}^{N} \left[ J(\theta_t + \epsilon_k) - J(\theta_t - \epsilon_k) \right] \epsilon_k$$

where $J(\theta)$ is the reward obtained by $\theta$, $\sigma_J^t$ is the standard deviation of all rewards obtained at time $t$ as proposed by Mania et al. (2018), and $\alpha_t$ are adaptive learning rates selected by ADAM (Kingma & Ba, 2014). The standard deviation $\sigma$ of the distribution that is generating the perturbations is treated as a hyperparameter.

We train NLPROLOG with ES using a learning from entailment setting (Muggleton & Raedt, 1994), in which the model is trained to decide whether a Prolog program $\mathcal{R}$ entails the truth of a candidate triple $c$. The objective of the model is to assign high probabilities $p(c|\mathcal{R}; \theta)$ to true candidate triples and low probabilities to false triples. To achieve this, we model the reward as $J(\theta) = yp(c|\mathcal{R}; \theta)$, where $y \in \{-1, 1\}$ is the gold label. To estimate $p(c|\mathcal{R}; \theta)$, we exhaustively search for all proofs for the triple $c$, up to a given depth $D$ which we treat as a hyperparameter. This search yields a number of proofs each with a success score $S_i$. We set $p(c|\mathcal{R}; \theta)$ to be the maximum of these scores $S_{\max} = \max_i S_i$.

### 4.3 GRADUAL RULE LEARNING

The reasoning process of NLPROLOG crucially depends on rules that describe the relations between predicates. While it is possible to write down rules in natural language, this approach is hardly scalable. Thus, we follow Rocktäschel & Riedel (2017) and use rule templates to perform Inductive Logic Programming (ILP) (Muggleton, 1991) which allows NLPROLOG to learn rules from training data. For this, a user has to define a set of rules with a given structure as input. Then, NLPROLOG randomly initializes the predicates of these rules. For instance, to induce a rule that can model transitivity, one would add a rule template of the form $p_1(X, Z) \Leftarrow p_2(X, Y) \wedge p_3(Y, Z)$, and NLPROLOG would instantiate multiple rules with randomly initialized embeddings for $p_1$, $p_2$, and $p_3$. The exact number and structure of the rule templates is treated as a hyper-parameter. If not explicitly stated otherwise, the experiments were performed with the same set of rule templates containing two rules for each of the forms $p_1(X, Y) \Leftarrow p_2(X, Y)$, $p_1(X, Y) \Leftarrow p_2(Y, X)$ and $p_1(X, Z) \Leftarrow p_2(X, Y) \wedge p_3(Y, Z)$.

In preliminary experiments, we observed that unification with such randomly initialized embeddings always leads to a stark drop of the proof success score. This implies in turn that proofs involving rule templates rarely yield the highest score $S_{\max}$ and thus, might have no impact on the value of the

reward function. Due to that, the expected gradient estimate for the rule embeddings is zero, and they will remain close to their initialization.

The main reason for this behavior is that the monotonicity of the aggregation functions implies that each sub-goal created by an application of a rule will only decrease the success score of the proof. Thus, all other things being equal, rules with a small number of body atoms (and facts in particular) will be preferred over more complex rules with a higher number of body atoms. Note that this problem is particularly severe in our setting where rules are initialized randomly while the remaining predicate symbols are instantiated using pretrained sentence embeddings.

We propose a Gradual Rule Learning (GRL) algorithm which counteracts this effect during training. GRL segments the training process into $B_{\max} + 1$ phases, where $B_{\max}$ is the maximum body size of all available rules. In the $k$-th phase of GRL, only proofs which employ at least one rule with a body size of $B_{\max} + 1 - k$ are considered to estimate $p(t|F; \theta)$. Thus, it is guaranteed that in each training step in phase $k$ at least one rule with a body size of $B_{\max} + 1 - k$ receives a training signal.

## 5 EVALUATION

We evaluate our method on two different QA data sets: BABI-1K-STATIC (Weston et al., 2015) and different subsets of WIKIHOP (Welbl et al., 2017). The used hyperparameter configurations can be found in Section B.

### 5.1 SUBSETS OF WIKIHOP

We evaluate on different subsets of WIKIHOP (Welbl et al., 2017), each containing a single query predicate. We consider the predicates *publisher*, *developer*, and *country*, because their semantics ensure that the annotated answer is unique and they contain a relatively high amount of questions that are annotated as requiring true multi-hop reasoning. For *publisher*, this yields 509 training and 54 validation questions, for *developer* 267 and 29, and for *country* 742 and 194. As the test set of WIKIHOP is not publicly available, and splitting the small train set would lead to a far too small validation set, we report scores for the validation set and refrain from hyperparameter optimization and early stopping.

Each data point consists of a query $p(q, X)$ where $q$ is some query entity, $X$ is the entity that has to be predicted, $C$ is a list of candidates entities, $a \in C$ is an answer entity and $p \in \{publisher, developer, country\}$ is the query predicate. In addition, every query is accompanied by a set of support documents which can be used to decide which candidate is the correct answer. To transform the support documents to natural language triples, we use the Open IE system MINIE (Gashteovski et al., 2017). We use the publicly available version of MINIE[3] in the dictionary mode, and use a list of all WIKIHOP candidate entities as our dictionary of multi-token entities.

Following Welbl et al. (2017), we use the two neural QA models BIDAF (Seo et al., 2016a) and FASTQA (Weissenborn et al., 2017) as baselines for the selected predicates of WIKIHOP. We employ the implementation provided by the QA framework JACK[4] (Weissenborn et al., 2018) with the same hyperparameters as used by Welbl et al. (2017) and train a separate model for each predicate. In order to compensate for the fact that both models are extractive QA models which cannot make use of the candidate entities, we additionally evaluate modified versions which transform both the predicted answer and all candidates to vectors using the *wiki-unigrams* model[5] of SENT2VEC (Pagliardini et al., 2017). Consequently, we return the candidate entity which has the highest cosine similarity to the predicted entity.

### 5.2 BABI-1K-STATIC

We utilize a subset of BABI-1K to study the behavior of NLPROLOG in a very controlled environment. Note however, that the experiments on WIKIHOP are much more significant, as they involve natural linguistic variability. BABI-1K-STATIC consists of the tasks QA4,QA15,QA17, and QA18 from

---

[3]`https://github.com/rgemulla/minie`
[4]`https://github.com/uclmr/jack`
[5]`https://drive.google.com/open?id=0B6VhzidiLvjSa19uYWlLUEkzX3c`

the BABI suite (Weston et al., 2015), each containing 1,000 training and 1,000 testing questions. These tasks were selected because, unlike the other tasks of BABI, they do not require reasoning about a dynamically changing world state, which is not supported by the current implementation of NLPROLOG. We automatically transform all statements and queries of the respective tasks to triples and use the resulting KB as input to NLPROLOG. We train and evaluate on each problem individually. The tasks QA4 and QA15 require entities as an output, thus we consider every entity that occurs at least once in any problem of the task as a candidate for all problems. Tasks QA17 and QA18 are binary classification tasks, and thus we determine the optimal threshold on the training set, after the training of NLPROLOG has finished.

We refrain from systematically comparing results on the individual BABI tasks to competing methods like Seo et al. (2016b); Peng et al. (2015); Dhingra et al. (2018); Henaff et al. (2016); Sukhbaatar et al. (2015), since our non-negligible preprocessing and evaluation on only four out of 20 tasks does not allow us to match the relevant evaluation protocols. We therefore utilize BABI-1K-STATIC only for ablation experiments, but note that NLPROLOG achieves similar or better accuracy values than the mentioned methods in all instances we studied, except on *QA4*.

## 5.3 DEALING WITH CANDIDATES

All questions of WIKIHOP and some of BABI-1K-STATIC include a set of answer candidates $C$. For those cases, we modify our reward function to leverage this information, taking inspiration from Bayesian Personalized Ranking (Rendle et al., 2009):

$$J(\theta) = p(a|\mathcal{R};\theta) - \max_{c \in C \setminus \{a\}} p(c|\mathcal{R};\theta),$$

where $a \in C$ is the true answer.

We observed that this reward function does not work well with the minimum aggregation function. Therefore, we employ this modified reward only when using the product aggregation and utilize the reward described in Section 4.2 with the minimum aggregation.

## 5.4 RESULTS

The results for the selected predicates of WIKIHOP can be found in Table 1. While there is no single best performing model, NLPROLOG is outperformed by at least one neural QA model on every predicate. For *country*, this only holds true when considering the versions of the neural models that have been augmented to consider candidate entities. For all three predicates, only a single transitive rule is utilized across all validation questions. Since we observe a more diverse set of induced rules on BABI-1K, we partly attributed this lack of diverse rules to the multi-hop characteristic of the WIKIHOP data. It seems that NLPROLOG struggles to find meaningful rules for the predicates *developer* and *publisher*, which leads to very few proofs involving rules on the development set: 1 out of 54 for *publisher* and 2 out of 29 for *developer*, compared with 159 out of 194 for *country*. We partially attribute this to the fact, that the semantics of *country* suggest a straightforward rule (transitivity of location), which is not true for *developer* or *publisher*. Additionally, the annotations regarding the neccessity of multi-hop reasoning provided for the validation set (Welbl et al., 2017) suggest that *publisher* and *developer* contain significantly fewer training examples involving multi-hop reasoning.

Exemplary proofs generated by NLPROLOG on the predicates *developer* and *country* can be found in Fig. 3. As we are especially interested in assessing the capability for multi-hop reasoning, we additionally evaluate on a subset of problems which have been unanimously labelled as requiring multi-hop reasoning. On this subset of the development data, which is denoted as *Countryhop (mh)* in Table 1, NLPROLOG outperforms all other single models.

## 5.5 NEURAL QA AND NLPROLOG ENSEMBLE

If the proof of NLPROLOG producing the prediction does not employ any rules, the prediction is essentially the result of performing a simple nearest neighbor search among the embeddings of all fact triples. We hypothesize that the neural QA models FASTQA and BIDAF are better suited for finding the most predictive single support statement, which motivates an ensemble of a neural QA model and NLPROLOG. We built a system which predicts the output of NLPROLOG when it used at

| Model | publisher | developer | country | country (mh) |
|---|---|---|---|---|
| BIDAF | 66.67 | 65.52 | 53.09 | 53.19 |
| FastQA | 62.96 | 62.07 | 57.21 | 57.44 |
| BIDAF-Sent2Vec | 75.93 | 68.97 | 61.86 | 65.96 |
| FastQA-Sent2Vec | 75.93 | 58.62 | 64.95 | 65.96 |
| NLProlog (ours) | 51.85 | 48.28 | 63.92 | 68.09 |
| BIDAF-Sent2Vec + NLProlog | **77.78** | **72.41** | 65.46 | **70.21** |
| FastQA-Sent2Vec + NLProlog | **77.78** | 62.07 | **69.07** | **70.21** |

Table 1: Accuracy scores in percent for different predicates on the development set of WIKIHOP. *country (mh)* only considers the problems that have been unanimously judged to require multi-hop reasoning. The upper part lists single models and the bottom part ensembles.

least one rule and the output of the neural QA model otherwise. This allows to employ the multihop reasoning of NLPROLOG when possible and to utilize the pattern matching capabilities of the neural QA models in the remaining cases. The results for these ensembles are given in the bottom part of Table 1. In all instances, ensembling a neural QA model with NLPROLOG improved upon all single models, indicating that they complement each other well. We analyze reasons for the success of this ensembling strategy in Section 5.6.

## 5.6 ERROR ANALYSIS

We conducted an extensive error analysis, in which we manually classified all errors of NLPROLOG on the selected WIKIHOP predicates into predefined categories. The results are depicted in Figure 2.

For the *developer* predicate, in the majority of cases, errors were caused by OPENIE during the fact extraction step: in all but one case, OPENIE did not produce the necessary facts, or a necessary facts was not stated in the support text, or there were multiple correct candidates and NLPROLOG selected the wrong one. As a consequence, for both the *publisher* and *developer* predicates, the majority of queries would not answerable, even when the necessary rules were correctly induced. The predictive accuracy was significantly higher for the *country* predicate, where errors were mostly due to entitities not having SENT2VEC embeddings and a few missing rules.

Fig. 2 indicates that FASTQA can produce the correct answer, even if crucial information is missing from the supporting documents. To analyze this further, we evaluated FASTQA and NLPROLOG on a test set of the *country* predicate in which all documents mentioning the query entity were removed. Remarkably, the accuracy of FASTQA *increased* by approximately $1\%$, while the accuracy of NLPROLOG decreased by approx. $11\%$.

Furthermore, we evaluated both FASTQA and NLPROLOG on the *hard* subset of *country* as defined by Sugawara et al. (2018): on these 62 problems which cannot be solved with a simple heuristic, NLPROLOG achieved an accuracy of $51.61\%$, as opposed to $46.77\%$ by FASTQA.

We conjecture that – besides NLPROLOG's multi-hop reasoning capability – this is one reason why the neural models and NLPROLOG complement each other nicely: neural models can compensate for missing information, while NLPROLOG is less susceptible for spurious correlations between the query and supporting texts. The complementary nature of both approaches is further supported by the error analysis, described in Fig. 4.

## 5.7 ABLATION EXPERIMENTS

We perform experiments on BABI-1K-STATIC to investigate the effect of the GRL training procedure. Specifically, we perform experiments on BABI-1K-STATIC with only the last phase of GRL (i.e. training without GRL), with the last and the penultimate phase, and with all three phases, corresponding to full GRL as we limit the rule complexity to two body atoms in rule templates. To maintain comparability between runs, we keep the number of SGD steps constant across all experiments.

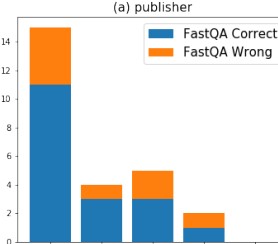 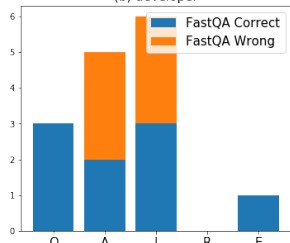 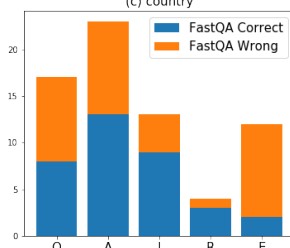

Figure 2: Results of the error analysis. The y-axis shows the number of errors per category: *O(penIE)* refers to a crucial information missing due to an error of the OpenIE tool, *A(mbiguous)* means that multiple candidates were correct and *NLProlog* chose one which was not labeled as the correct one, while *I(nfo)* refers to problems that are not answerable with the provided support texts. *R(ule)* means that a required rule was not induced, and *E(mbedding)* implies that the answer was correctly deduced but received a lower score than an erroneous deduction of another candidate.

|  | QA4 | QA15 | QA17 | QA18 |
|---|---|---|---|---|
| GRL phase 3 (no) | 56.4 | 37.1 | 62.8 | 90.9 |
| GRL phase 2+3 | **88.8** | 33.5 | 66.7 | 94.7 |
| GRL phase 1+2+3 (full) | 88.0 | **100.0** | **67.6** | **97.6** |
| Gold Rules (Untrained) | **100.0** | **100.0** | 73.2 | 92.0 |
| Gold Rules (Trained) | **100.0** | **100.0** | **84.8** | **98.1** |
| No Rules (Trained) | 60.0 | 32.8 | 63.0 | 90.4 |
| Ten Templates | 88.1 | **100.0** | 61.9 | 89.4 |

Table 2: Accuracies in percent of ablation experiments on BABI-1K-STATIC. The upper half examines the effectiveness of GRL, while the bottom results concern the effect of training and different choices of rules.

Additionally, we experiment with manually defined rules, which we deem sufficient for solving each of the four tasks. For these, we report results before and after training, as well as for a run without any rule templates. The accuracy scores for all experiments on BABI are provided in Table 2.

To assess the impact of the choice of rule templates, we evaluate NLPROLOG on *bAbI-1k-static* with a different set of rule templates containing two rules of the form $p_1(X, Y) \Leftarrow p_2(X, Y)$, four with the form $p_1(X, Y) \Leftarrow p_2(Y, X)$ and another four for $p_1(X, Z) \Leftarrow p_2(X, Y) \wedge p_3(Y, Z)$.

Clearly, the full GRL is necessary to obtain acceptable results in any of the problems. Interestingly, phase 1 of GRL does not contribute anything for QA4, which is perfectly solvable using only rules of body size 1. On the other hand, QA15 and QA18 both require a rule of body size 2, which makes phase 1 strongly improve the results. Only for $QA17$ the results are inconclusive. Nevertheless, this indicates that GRL works as intended, with the earlier phases encouraging the induction of rules with a higher number of conjuncts in the body.

The results using manually defined rules suggest that even when sufficient rules are provided, training with ES is helpful nevertheless. Interestingly, the model using no rules is able to solve over 90% of the problems in *QA18*, indicating that this problem is not well suited for evaluating reasoning capabilities of QA models.

Using ten instead of six templates leads to worse performance on all BABI problems but *QA15*, which is solved perfectly with a much faster convergence rate. This result indicates that the choice of rule templates might be an important hyperparameter which should be optimized for a given problem.

## 6 DISCUSSION AND FUTURE WORK

We have developed NLPROLOG, a system that is able to perform rule-based reasoning on natural language input, and can learn domain-specific natural language rules from training data. To this end,

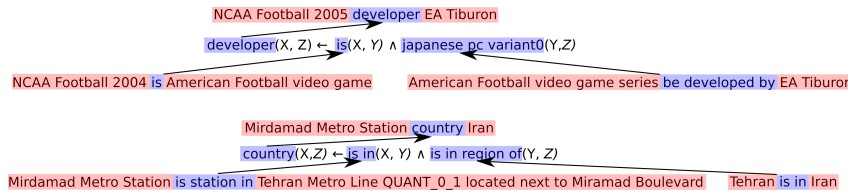

Figure 3: Example proof trees generated by NLPROLOG. Each of the two trees shows an application of a transitive rule, the first for the predicate *developer* and the second for the predicate *country*. The rule templates are displayed with the most similar predicate. Note the noise introduced by the Open IE process, e.g. *QUANT_0_1* and that entities and predicates do not need to match exactly.

we have proposed to combine a symbolic prover with pretrained sentence embeddings and to train the resulting system with Evolution Strategies. We have evaluated NLPROLOG on two different QA tasks, showing that it can learn domain-specific rules and produce predictions which complement those of the two strong baselines BIDAF and FASTQA. This allows to build an ensemble of a baseline and NLPROLOG which outperforms all single models.

While we have focused on a subset of First Order Logic in this work, the expressiveness of NL-PROLOG could be extended by incorporating a different symbolic prover. For instance, a prover for temporal logic (Orgun & Ma, 1994) would allow to model temporal dynamics in natural language and enable us to evaluate NLPROLOG on the full set of BABI tasks. We are also interested in incorporating future improvements of symbolic provers, Open IE systems and pretrained sentence representations to further enhance the performance of NLPROLOG. To study the performance of the proposed method without the noise introduced by the Open IE step, it would be useful to evaluate it on tasks like knowledge graph reasoning. Additionally, it would be interesting to study the behavior of NLPROLOG in the presence of multiple WIKIHOP query predicates.

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

# A    ALGORITHMS

## A.1    WEAK UNIFICATION

**fun** *unify*$(x, y, \theta, S)$
 **Input:**
 $x$: function $f(\ldots)$ | atom $p(\ldots)$ | *variable* | list $x_1 :: x_2 :: \ldots :: x_n$
 $y$: function $f'(\ldots)$ | atom $p'(\ldots)$ | *variable* | list $y_1 :: y_2 :: \ldots :: y_m$
 $\theta$: current substitutions, default = $\{\}$
 $S$: current success score, default = $1.0$
 **Output:** (Unifying substitution $\theta'$ or *failure*, Updated success score $S'$)
 **if** $\theta = $ *failure* **then return** (*failure*, $0$)
 **else if** $S < \lambda$ **then return** (*failure*, $0$)
 **else if** $x = y$ **then return** ($\theta$, $S$)
 **else if** $x$ *is Var* **then return** *unify_var*$(x, y, \theta, S)$
 **else if** $y$ *is Var* **then return** *unify_var*$(y, x, \theta, S)$
 **else if** $x$ *is* $f(x_1, \ldots, x_n)$, $y$ *is* $f'(y_1, \ldots, y_n)$*, and* $f \sim f' \geq \lambda$ **then**
  $S' := S \wedge f \sim f'$
  **return** *unify*$(x_1 :: \ldots :: x_n, y_1 :: \ldots :: y_n, \theta, S')$
 **end**
 **else if** $x$ *is* $p(x_1, \ldots, x_n)$, $y$ *is* $p'(y_1, \ldots, y_n)$*, and* $p \sim p' \geq \lambda$ **then**
  $S' := S \wedge f \sim f'$
  **return** *unify*$(x_1 :: \ldots :: x_n, y_1 :: \ldots :: y_n, \theta, S')$
 **end**
 **else if** $x$ *is* $x_1 :: \ldots :: x_n$ *and* $y$ *is* $y_1 :: \ldots :: y_n$ **then**
  $(\theta', S') := $ *unify*$(x_1, y_1, \theta, S)$
  **return** *unify*$(x_2 :: \ldots :: x_n, y_2 :: \ldots :: y_n, \theta', S')$
 **end**
 **else if** $x$ *is empty list and* $y$ *is empty list* **then return** ($\theta$, $S$)
 **else return** (*failure*, $0$)
**fun** *unify_var*$(v, o, \theta, S)$
 **if** $\{v/val\} \in \theta$ **then return** *unify*$(val, o, \theta, S)$
 **else if** $\{o/val\} \in \theta$ **then return** *unify*$(var, val, \theta, S)$
 **else return** $(\{v/o\} + \theta, S)$

**Algorithm 1:** The weak unification algorithm in Spyrolog without occurs check

## A.2    RUNTIME OF PROOF SEARCH

The worst case complexity vanilla logic programming is exponential in the depth of the proof (Russell & Norvig, 2010). However, in our case this is a particular problem because weak unification requires the prover to attempt unification between all entity/predicate symbols.

To keep things tractable, NLPROLOG only attempts to unify symbols with a similarity greater than some user-defined threshold $\lambda$. Furthermore, in the search step for one statement $q$, for the rest of the search, $\lambda$ is set to $\lambda := \max(\lambda, S)$ whenever a proof for $q$ with success score $S$ is found. Due to the monotonicity of the employed aggregation functions, this allows to prune the search tree without losing the guarantee to find the proof yielding the maximum success score. We found this optimization to be crucial to make the proof search scale for the studied *wikihop* predicates.

# B    HYPERPARAMETER CONFIGURATION

On BABI-1K we optimize the embeddings of predicate symbols of rules and query triples, as well as of entities. WIKIHOP has a large number of unique entity symbols and thus, optimizing their embeddings is prohibitive. Thus, we only train the predicate symbols of rules and query triples on this data set. The embeddings for entities and predicate symbols of fact and query triples are initialized using the WIKI-UNIGRAMS model of SENT2VEC, while the embeddings of rule predicates are intialized by uniformly sampling from the interval $[-\frac{1}{\sqrt{600}}, \frac{1}{\sqrt{600}}]$. All experiments were performed

with the same set of rule templates containing two rules for each of the forms $p(X, Y) \Leftarrow q(X, Y)$, $p(X, Y) \Leftarrow q(Y, X)$ and $p(X, Z) \Leftarrow q(X, Y) \land r(Y, Z)$ and set the similarity threshold $\lambda$ to $0.3$. At each optimization step, we evaluate 100 perturbations sampled from $\mathcal{N}(0, 0.04)$ on a mini-batch of 16 training problems and use all of the directions in the generation of the next weight vector. If not stated otherwise, we use GRL with three phases training for 500 mini-batches in each phase. For the predicates *publisher* and *developer*, we used 1,000 mini-batches in the final phase of GRL. To further encourage rule usage, we use the *minimum* aggregation function in all but the last phases of GRL, in which we switch to *product*.

## C   ERROR SETS

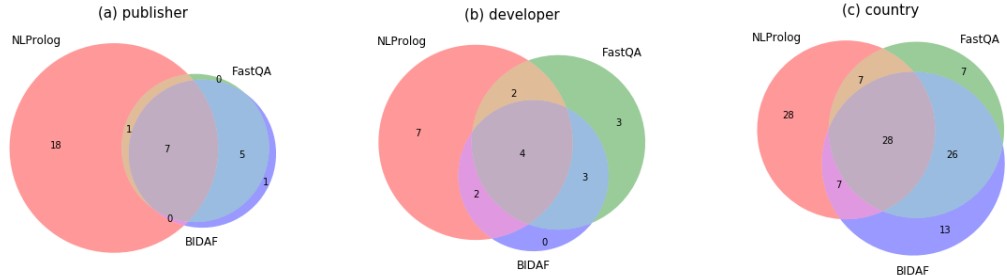

Figure 4: Venn diagram of the error sets per predicate.

