# OpenReview forum: "NLProlog: Reasoning with Weak Unification for Natural Language Question Answering"
_ICLR.cc/2019/Conference_

### Official Review · AnonReviewer1 · 2018-11-01
**Promising unification of neural representations with learned logical rules**

**Rating:** 7
**Confidence:** 4

**Review:**



UPDATE: Given the authors' rebuttal and the clear improvements to their paper, I've increased my rating of the work.

=======================

This paper presents NLProlog, a method that combines logical rules with distributed representations for reasoning on natural language statements. Natural language statements (first converted to logical triples) and templated logical rules are embedded in a vector space using a pretrained sentence encoder. These embedded "symbols" can be compared in vector space, and their similarity used in a theorem prover (Prolog) modified to support weak unification. The theorem prover determines the answer to a natural language query by constructing a proof according to its logical rules.

Training through the non-differentiable theorem prover occurs via an "evolutionary strategy," which enables the model to fine-tune its sentence encoders and learn domain-specific logic rules directly from text. The authors also propose a Gradual Rule Learning (GRL) algorithm that seems necessary for the optimization process to converge on good solutions.

Despite the model's complexity, the paper was fairly clear to me.

Although the proposed model is a conglomeration of pre-existing parts, the combination is original to my knowledge. The use of Open Information Extraction to transform natural language statements to logical statements, which amenable to theorem provers, is novel and also circumvents the complicated preprocessing required by previous related works.

The authors evaluate the proposed approach on subsets of the Wikihop dataset and BABI-1k. NLProlog performs competitively with neural models, similarly augmented with Sent2Vec but lacking explicit logical rules, only on the 'country' subset of Wikihop. It does not compete with or clearly outperform these models in general. As the authors state, it further "struggles to find meaningful rules for the predicates 'developer' and 'publisher'."

NLProlog demonstrates strong performance on a subset of problems labelled unanimously by annotators to require multi-hop reasoning. Unfortunately, this is only done for the "country" subset of Wikihop, on which the model was already shown to have the strongest performance. I'd find this more convincing if similar improvements were shown on the other subsets (publisher, developer).

Taking into account also that the BABI subset was used only for ablation, the limited results call into question the significance of the work. It would definitely benefit from more extensive experimental validation. On the other hand, it's very positive to see that NLProlog seems to succeed where the neural models fail, and vice versa, so that the two approaches can be combined in an ensemble to achieve state-of-the-art results. This suggests that the paper's line of research has something to add to the community and should be pursued further. I'd find this result more interesting if an error analysis elucidated some characteristics of the examples that each approach does well/poorly on.

I'd like to see more analysis in general, that answers questions including:
- How reliable is the Open IE system and how does its performance impact the end task?
- How well-specified must the a priori rule structures be to achieve good performance? Further, how does the number and structure of the rules (a hyperparameter in this work) affect performance?
- What is the run-time/complexity of the exhaustive proof search during training?
- Relatedly, you state that you limit the rule complexity to two body atoms in the rule templates for BABI. Can you estimate what rule complexity is required in the Wikihop tasks?

I would like to recommend this work more confidently because it tackles such an important problem and does so in an interesting, well-conceived way. My reluctance arises from the limited experimental validation and analysis. Given more analysis details and experimental evidence from the authors, I'm happy to raise my recommendation.

Pros:
- the method complements standard deep QA models to achieve state-of-the-art results in an ensemble.
- unifying neural representations with logical/symbolic formalisms is an important research direction.
- a code release is planned.

Cons:
- a very complex model, whose details are occasionally unclear; the algorithms in Appendix A are helpful but they are not in the main text.
- the model expresses only a limited subset of first order logic; dynamically changing world states are not supported (yet).
- limited experimental validation.
- it's good to be able to incorporate prior knowledge, but it seems like it's quite necessary to pre-specify rules (in template form).

Minor quibble: Evolutionary learning strategies, such as genetic algorithms, go back a long way. It's strange using only a reference from 2017 to introduce them.

---

> ### Author Response · Authors · 2018-11-23
> **Evaluation and error analysis**
>
> Thanks a million for your detailed feedback and suggestions. We are glad to hear that you found our paper to be clear and this line of work interesting, particularly the fact that NLProlog succeeds where existing neural models fail.
>
> Q1. Unfortunately, this is only done for the "country" subset of WikiHop, on which the model was already shown to have the strongest performance. I'd find this more convincing if similar improvements were shown on the other subsets (publisher, developer).
>
> Extending our evaluation with further relationships would certainly be a worthwhile extension of our paper. Following your advice, we performed additional experiments on the predicate record_label. In these experiments, OpenIE was not able to extract the relevant facts, so it was not possible to apply our framework. Following Q3 (below), we performed an extensive error analysis to also explaining results on “developer” and “publisher” questions (see Section 5.6).
>
> Q2. On the other hand, it's very positive to see that NLProlog seems to succeed where the neural models fail, and vice versa, so that the two approaches can be combined in an ensemble to achieve state-of-the-art results.
>
> Thanks for your comment. Indeed, the last two rows of Table 1 show consistently strong results for an ensemble of the proposed logic-based approach and deep neural models. We rephrased Section 5.5 to better explain our ensembling strategy and added a detailed qualitative analysis (see Q3 below) demonstrating that the two approaches have complementary strengths.
>
> Q3. I'd find this result more interesting if an error analysis elucidated some characteristics of the examples that each approach does well/poorly on.
>
> Thank you for this excellent suggestion. We added to the paper an extensive error analysis, see Section 5.6. The takeaway message is that (up to issues with data quality) the OpenIE component causes the majority of NLProlog errors, while neural Q&A models are robust even when crucial information is missing in the support documents.  However, neural Q&A models often also pick up spurious data artifacts which sometimes coincidentally even lead to correct results (see Section 5.6). We show that NLProlog is much less affected by such effects. Furthermore, NLProlog provides proofs for its predictions which make it easy for users to spot errors.
>
> Q4. How well-specified must an priori rule structure be to achieve good performance? Further, how does the number and structure of the rules (a hyperparameter in this work) affect performance?
>
> Please note that we use the same rule templates in all evaluation results - we apologize that this was mentioned only in the appendix in our submission. Actually, the model can learn more complex rules by composing simpler rules [1]. We employ two templates for each of the following three structures ‘p(X, Y) :- q(X, Y)’, ‘p(X, Y) :- q(Y, X)’, and ‘p(X,Z) :- q(X,Y), r(Y,Z)’, because these capture the basic reasoning steps of entailment, symmetry and transitivity (see Q12 regarding the appropriateness of this term). We studied the impact of having more/less structures in the ablation study on bAbI and found that these have a significant impact on convergence speed and overall performance. We expand on this in Section 5.7 of the revised paper.
>
> [1] Evans and Grefenstette, Learning Explanatory Rules from Noisy Data, 2017

---

> ### Author Response · Authors · 2018-11-23
> **Rules and run-time complexity**
>
> Q5. What is the run-time/complexity of the exhaustive proof search during training?
>
> As in most logic programming frameworks, the number of candidate proofs essentially grows exponentially with the depth of the proof trees. This is a particular problem in our setting where in principle all predicates match with all others (to a certain degree). However, we do not use an exhaustive proof search but a simple pruning heuristic which disregards all proof steps with a score lower than a given threshold. This threshold is updated dynamically during the search step. We now make this part of our method more clear in the appendix (see Section A.2).
>
> Q6. Relatedly, you state that you limit the rule complexity to two body atoms in the rule templates for bAbI. Can you estimate what rule complexity is required in the Wikihop tasks?
>
> As mentioned above, we hypothesized that direct entailment, symmetry, and transitivity are the most important types of reasoning. In our analysis we found that, for answering most questions on the considered WikiHop predicates, a limited number of rules of the form  ‘p(X,Z) :- p(X,Y), p(Y,Z)’ is sufficient. This is because the WikiHop dataset was constructed by traversing the graph connecting the entities mentioned in the support texts, a property that can be well exploited by our method.
>
> Q7. Minor quibble: Evolutionary learning strategies, such as genetic algorithms, go back a long way. It's strange using only a reference from 2017 to introduce them.
>
> Thank you for pointing this out. We updated our references to better reflect the long tradition in studying evolutionary learning (See Section 1). Any further suggestions are welcome.

---

### Official Review · AnonReviewer2 · 2018-11-03
**Interesting direction, experiments are unconvincing**

**Rating:** 5
**Confidence:** 3

**Review:**

Updated after reading author revisions:
I appreciate the clarifications, the response answered almost all of my small technical questions.  That plus the new error analysis increases my opinion about the paper, and I'm no longer concerned that the rule templates are hand-generated given their generality and small number.  I am still concerned that we don't actually know how well the methods work, because the test sets are small and the performance differences between the methods (in Table 1) are quite close.  I will raise my score one point.

The authors might try to evaluate using k-fold cross-validation with the training set, to obtain more examples for evaluation.

Original review:

The paper presents a technique for using prolog along with neural representations and Open IE to perform reasoning with weak unification.

I like the basic direction of trying to combine prolog with neural models, and the weak unification notion.  The approach seems sufficiently novel, and the GRL is a reasonable heuristic.

I do, however, have significant concerns about the experiments.  The data sets are selected subsets of other standard benchmarks, rather than the entire benchmarks, and the test sets are quite small (e.g., the "developer" column where the NLProlog approach shows some of the larger wins -- when ensembled with previous techniques -- is based on a test set of only 29 examples).  Given the hand-annotated nature of much of the input knowledge (the rule templates), this introduces an important concern that the experimental wins will not be robust in more realistic settings where different knowledge may be required.

Minor comments/questions
page 2: "without the need to transforming"
I did not understand how individual symbols, predicates and entities, have embeddings that come from sentence vectors (Section 4.1).
The learning objective in Section 4.2 seems reasonable, but I did not understand how "evolution" was part of the strategy there.
The example rule template for transitivity isn't actually transivity unless p_i=p_j for all i,j, I found that a little confusing.
Where are "t-norms" (mentioned at the top of page 6) used?  I did not see this.
"candidates entities" -> "candidate entities"

---

> ### Author Response · Authors · 2018-11-23
> **Datasets and rule annotations**
>
> Thank you a lot for your feedback!
>
>
> Q8. The data sets are selected subsets of other standard benchmarks, rather than the entire benchmarks, and the test sets are quite small
>
> We agree that a more extensive evaluation would strengthen our work. However, we also note that NLProlog is a technique specifically designed to support multi-hop reasoning (see also Q2 above). As this today is a non-standard feature, most evaluation datasets we are aware of do not contain (or just a few) structured predicates which require such capabilities. For all performed evaluations in which our framework was applicable, we observe consistent improvements of NLProlog when ensembled with neural Q&A models. Following Q3 (above) and Q14 (below), we added to the paper a detailed error analysis demonstrating that NLProlog has strengths that are directly complementary to neural Q&A models.
>
> Q9. Given the hand-annotated nature of much of the input knowledge (the rule templates), this introduces an important concern that the experimental wins will not be robust in more realistic settings where different knowledge may be required.
>
> Our method indeed requires the manual specification of rule templates. However, we actually use the same set of just six rule templates across all tasks (minus the ablation study). Rule instantiation as required by a specific task is performed automatically during the learning phase. We rephrased Section 4.3 to make this difference more clear.
>
> Whether the proposed approach would work if multiple query predicates are involved indeed is an open yet very interesting question. We added this thought as future work to Section 6.
>
> Q10. I did not understand how individual symbols, predicates and entities, have embeddings that come from sentence vectors (Section 4.1)
>
> We associate every entity and predicate to an embedding vector, initialised with the Sent2Vec sentence encoder: starting from text, we extract the relevant facts via OpenIE, and encode their predicate and entities using Sent2Vec. We apologise if this was not clear enough and rephrased the description in Section 4.1.
>
> Q11. The learning objective in Section 4.2 seems reasonable, but I did not understand how "evolution" was part of the strategy there.
>
> “Evolution Strategies” is a gradient estimation method proposed in [1], which is commonly also interpreted under an evolutionary computing perspective [2]. In this case, “Evolution” stems from the fact that the gradient is a function of a population of sampled model parameters. We rephrased Section 1 to make these issues clearer.
>
> [1] Salimans et al., Evolution Strategies as a Scalable Alternative to Reinforcement Learning, 2017
> [2] Eiben, Agoston E., and James E. Smith. Introduction to evolutionary computing. Vol. 53. Berlin: springer, 2003.
>
> Q12. The example rule template for transitivity isn't actually transitivity unless p_i=p_j for all i,j, I found that a little confusing.
>
> Thank you for pointing this out—indeed “transitivity” is not the best term here. We changed it to “multi-hop rule” throughout the paper.
>
> Q13. Where are "t-norms" (mentioned at the top of page 6) used? I did not see this.
>
> Sorry, this was a typo from an older version of the manuscript. While our aggregation functions are t-norms in a mathematical sense [3], we have replaced the mention by “aggregation function” to be consistent with the rest of the paper.
>
> [3] Sessa, Maria I. 2002. “Approximate Reasoning by Similarity-Based SLD Resolution.” Theoretical Computer Science 275 (1): 389–426.

---

### Official Review · AnonReviewer3 · 2018-11-06
**interesting approach towards combining neural networks with logic reasoning**

**Rating:** 7
**Confidence:** 3

**Review:**

Update:
I appreciate the through error analysis the authors have done in the revision, which addressed my major previous concerns. I've updated my score accordingly.

This paper presents an approach to combine Prolog-like reasoning with distributional semantics. First, extracted fact triples are unified (i.e. mapped to) predicates and entities. Next, reasoning is performed with rule templates, where predicates and entities are abstracted. Since the reasoning process is non-differentiable, zero-oder optimization is used to fine-tune the predicate / entity embeddings.

The general idea of combining logical reasoning with neural models is quite appealing. A sketch of the algorithm is to first build structured knowledge from the text, then do reasoning over it to answer queries. In this work, the first step is completely relied on an off-the-shelf tool, Open-IE. It would be useful to see whether this step is the bottle neck of such approaches. One possibility is to apply the model to knowledge graph reasoning, which would remove any noise introduced from the knowledge extraction step, and solely focus on evaluating reasoning.

The results are a bit restricted, as in only a subset of the datasets are evaluated. I suspect part of the reason is that most of the QA datasets which claims to require multi-step reasoning don't really need much reasoning... However, it would be useful to do some simple (perhaps qualitative) analysis on the data quality, and make sure that it indeed tests what it intended to. For the ensemble results in Table 1, usually even ensembling same models trained with different seed would show improvements, so I'm not completely convinced that BiDAF and NLProlog are complementary - would be nice to see error analysis here.

Question:
What is the size of hand-coded predicates and rules? What's the coverage of these rules on the datasets, i.e. are there questions unanswerable by the provided rules?

Overall, while the results are limited, the approach is interesting, and hopefully will spur more work towards interpretable models with explicit reasoning.

---

> ### Author Response · Authors · 2018-11-23
> **Hybrid NLProlog-Neural models, OpenIE, and rules**
>
> Thank you for your feedback and suggestions. It is great to hear that you found our paper interesting.
>
> Q14. In this work, the first step is completely relied on an off-the-shelf tool, Open-IE. It would be useful to see whether this step is the bottleneck of such approaches.
>
> Thank you for this useful suggestion. We performed an in-depth error analysis of NLProlog which revealed that indeed a large portion of errors are due to a failure in the OpenIE step (see Q3). We added this analysis to Section 5.6.
>
> Q15. For the ensemble results in Table 1, usually even ensembling same models trained with different seed would show improvements, so I'm not completely convinced that BiDAF and NLProlog are complementary - would be nice to see error analysis here.
>
> This is an important observation and question. Indeed, the error analysis we now added to the paper supports our hypothesis that the two approaches are complementary on our evaluation data (see Section 5.6.). The proposed method and neural Q&A models seem to have orthogonal inductive bias, and thus complementary strengths and weaknesses.
>
> Q16. Question: What is the size of hand-coded predicates and rules?
>
> Please also see Q4, Q6, and Q9: in all experiments, we use the same set of just six rule templates, since rules with two body atoms are sufficient for expressing arbitrarily complex rules [1].
>
> [1] Evans and Grefenstette. Learning Explanatory Rules from Noisy Data. JAIR 2018
>
> Q17. What is the coverage of these rules on the datasets, i.e. are there questions unanswerable by the provided rules?
>
> For “country” questions, the six rule templates in principle were sufficient to learn rules that can answer all problems but four -- see the new error analysis for the reasons where the actual results deviate from this statement. As stated in the paper, rule induction did not perform well for “developer” and “publisher” questions, either because OpenIE was not able to extract relevant facts (in the case of “publisher”), or a low proportion of answerable multi-hop questions (in the case of “developer”). For more detail, please see Section 5.6. We think that, given the correct rules, the provided rule templates would be sufficient for answering all studied queries. See Q6 for an additional discussion of this issue.

---

### Author Response · Authors · 2018-11-23
**In-depth error analysis, additional experiments, extensive improvements in clarity**

Summary of changes, Nov 23, 2018
We thank all three reviewers for their detailed and insightful feedback. We used it to update our submission by introducing the following changes:

- We added an extensive error analysis (Section 5.6) which elucidates the strengths and weaknesses of NLProlog and the neural QA models and provides additional evidence that the approaches are indeed complementary. Additionally, the error analysis revealed that the OpenIE step is a likely bottle neck of NLProlog, which shows a path for future improvement.

- We updated Section 5.7 with an additional experiment on bAbI which studies the effect of varying the size of the rule templates, finding that the number and structure of rules has a strong effect on the convergence speed and predictive performance.

- We updated various sections to improve the clarity of the paper, especially with regards to the details of the proposed method: we clarified the structure of the employed rule templates in Section 4.3, added Section A.2 to discuss the runtime of the proof search step and applied optimizations, as well as expanded on the initialization of the symbol embeddings in Section 4.1.

---

### Meta-Review · Area_Chair1 · 2018-12-18
**Important contribution, but ultimately weak evaluation**

**Confidence:** 3
**Recommendation:** Reject

**Metareview:**

This paper combines Prolog-like reasoning with distributional semantics, applied to natural language question answering. Given the importance of combining neural and symbolic techniques, this paper provides an important contribution. Further, the proposed method complements standard QA models as it can be easily combined with them.

The reviewers and AC note the following potential weaknesses:
(1) The evaluation consisted primarily on small subsets of existing benchmarks,
(2) the reviewers were concerned that the handcrafted rules were introducing domain information into the model, and (3) were unconvinced that the benefits of the proposed approach were actually complementary to existing neural models.

The authors addressed a number of these concerns in the response and their revision. They discussed how OpenIE affects the performance, and other questions the reviewers had. Further, they clarified that the rule templates are really high-level/generic and not "prior knowledge" as the reviewers had initially assumed. The revision also provided more error analysis, and heavily edited the paper for clarity. Although these changes increased the reviewer scores, a critical concern still remains: the evaluation is not performed on the complete question-answering benchmark, but on small subsets of the data, and the benefits are not significant. This makes the evaluation quite weak, and the authors are encouraged to identify appropriate evaluation benchmarks.

There is disagreement in the reviewer scores; even though all of them identified the weak evaluation as a concern, some are more forgiving than others, partly due to the other improvements made to the paper. The AC, however, agrees with reviewer 2 that the empirical results need to be sound for this paper to have an impact, and thus is recommending a rejection. Please note that paper was incredibly close to an acceptance, but identifying appropriate benchmarks will make the paper much stronger.